# LONG-TAILED RECOGNITION BY ROUTING DIVERSE DISTRIBUTION-AWARE EXPERTS

**Xudong Wang[1], Long Lian[1], Zhongqi Miao[1], Ziwei Liu[2], Stella X. Yu[1]**
[1]UC Berkeley / ICSI, [2]Nanyang Technological University
{xdwang,longlian,zhongqi.miao,stellayu}@berkeley.edu
ziwei.liu@ntu.edu.sg

## ABSTRACT

Natural data are often long-tail distributed over semantic classes. Existing recognition methods tackle this imbalanced classification by placing more emphasis on the tail data, through class re-balancing/re-weighting or ensembling over different data groups, resulting in increased tail accuracies but reduced head accuracies.

We take a dynamic view of the training data and provide a principled model bias and variance analysis as the training data fluctuates: Existing long-tail classifiers invariably increase the model variance and the head-tail model bias gap remains large, due to more and larger confusion with hard negatives for the tail.

We propose a new long-tailed classifier called RoutIng Diverse Experts (RIDE). It reduces the model variance with multiple experts, reduces the model bias with a distribution-aware diversity loss, reduces the computational cost with a dynamic expert routing module. RIDE outperforms the state-of-the-art by 5% to 7% on CIFAR100-LT, ImageNet-LT and iNaturalist 2018 benchmarks. It is also a universal framework that is applicable to various backbone networks, long-tailed algorithms, and training mechanisms for consistent performance gains. Our code is available at: https://github.com/frank-xwang/RIDE-LongTailRecognition.

## 1 INTRODUCTION

Real-world data are often long-tail distributed over semantic classes: A few classes contain many instances, whereas most classes contain only a few instances. Long-tailed recognition is challenging, as it needs to handle not only a multitude of small-data learning problems on the tail classes, but also extreme imbalanced classification over all the classes.

There are two ways to prevent the many head instances from overwhelming the few tail instances in the classifier training objective: **1) class re-balancing/re-weighting** which gives more importance to tail instances (Cao et al., 2019; Kang et al., 2020; Liu et al., 2019), **2) ensembling over different data distributions** which re-organizes long-tailed data into groups, trains a model per group, and then combines individual models in a multi-expert framework (Zhou et al., 2020; Xiang et al., 2020).

We compare three state-of-the-art (SOTA) long-tail classifiers against the standard cross-entropy (CE) classifier: cRT and $\tau$-norm (Kang et al., 2020) which adopt a two-stage optimization, first representation learning and then classification learning, and LDAM (Cao et al., 2019), which is trained end-to-end with a marginal loss. In terms of the classification accuracy, a common metric for model selection on a *fixed* training set, Fig. 1a shows that, all these existing long-tail methods increase the overall, medium- and few-shot accuracies over CE, but *decrease the many-shot accuracy*.

These intuitive solutions and their experimental results seem to suggest that there is a head-tail performance trade-off in long-tailed recognition. We need a principled performance analysis approach that could shed light on such a limitation *if it exists* and provide guidance on how to overcome it.

Our insight comes from a dynamic view of the training set: It is merely a sample set of some underlying data distribution. Instead of evaluating how a long-tailed classifier performs on the fixed training set, we evaluate how it performs as the training set fluctuates according to the data distribution.

|  | All | | | Many-shot | | | Med-shot | | | Few-shot | | |
|---|---|---|---|---|---|---|---|---|---|---|---|---|
|  | acc | bias | var | acc | bias | var | acc | bias | var | acc | bias | var |
| CE | 31.6 | 0.60 | 0.47 | 57.3 | 0.28 | 0.35 | 28.2 | 0.61 | 0.51 | 6.3 | 0.94 | 0.57 |
| $\tau$-norm | 35.8 | 0.52 | 0.49 | 55.9 | 0.28 | 0.37 | 33.2 | 0.53 | 0.52 | 16.1 | 0.78 | 0.60 |
| cRT | 36.4 | **0.50** | 0.50 | 51.3 | 0.32 | 0.41 | 38.6 | **0.44** | 0.50 | 17.0 | 0.76 | 0.61 |
| LDAM | 34.4 | 0.53 | 0.51 | 55.1 | 0.28 | 0.38 | 31.9 | 0.53 | 0.54 | 13.9 | 0.81 | 0.63 |
| RIDE + LDAM | **40.5** | **0.50** | **0.42** | **60.5** | 0.28 | **0.30** | **38.7** | 0.50 | **0.44** | **20.1** | **0.74** | **0.52** |

(a) Comparisons of the mean accuracy, per-class bias and variance of baselines and our RIDE method. Better (worse) metrics than the distribution-unaware cross entropy (CE) reference are marked in green (red).

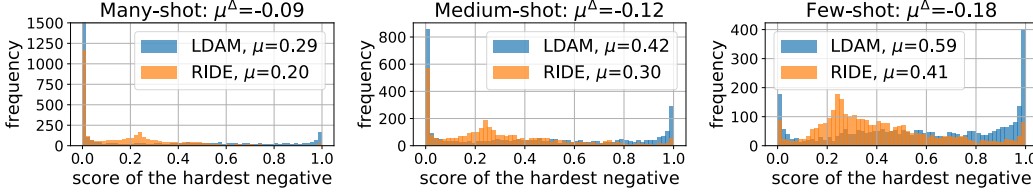

(b) Histograms of the largest softmax score of the other classes (the hardest negative) per instance.

Figure 1: Our method RIDE outperforms SOTA by reducing both model bias and variance. **a)** These metrics are evaluated over 20 independently trained models, each on a random sampled set of CIFAR100 with an imbalance ratio of 100 and 300 samples for class 0. Compared to the standard CE classifier, existing SOTA methods almost always increase the variance and some reduce the tail bias at the cost of increasing the head bias. **b)** The metrics are evaluated over CIFAR100-LT Liu et al. (2019). LDAM is more likely to confuse the tail (rather than head) classes with the hardest negative class, with an average score of 0.59. RIDE with LDAM can greatly reduce the confusion with the nearest negative class, especially for samples from the few-shot categories.

Consider the training data $D$ as a random variable. The prediction error of model $h$ on instance $x$ with output $Y$ varies with the realization of $D$. The expected variance with respect to *variable D* has a well-known bias-variance decomposition:

$$\text{Error}(x; h) = E[(h(x; D) - Y)^2] = \text{Bias}(x; h) + \text{Variance}(x; h) + \text{irreducible error}(x). \quad (1)$$

For the above L2 loss on regression $h(x) \to Y$, the model bias measures the *accuracy* of the prediction with respect to the true value, the variance measures the *stability* of the prediction, and the irreducible error measures the *precision* of the prediction and is irrelevant to the model $h$.

Empirically, for $n$ random sample sets of data, $D^{(1)}, \ldots, D^{(n)}$, the $k$-th model trained on $D^{(k)}$ predicts $y^{(k)}$ on instance $x$, and collectively they have a mean prediction $y_m$. For the L2 regression loss, the model bias is simply the L2 loss between $y_m$ and ground-truth $t = E[Y]$, whereas the model variance is the variance of $y^{(k)}$ with respect to their mean $y_m$:

$$\text{L2 regression loss:} \qquad \mathcal{L}(y; z) = (y - z)^2 \qquad\qquad\qquad (2)$$

$$\text{mean prediction:} \qquad y_m = \frac{1}{n}\sum_{k=1}^{n} y^{(k)} \qquad\qquad = \arg\min_z E_D[\mathcal{L}(h(x); z)] \quad (3)$$

$$\text{model bias:} \qquad \text{Bias}(x; h) = (y_m - t)^2 \qquad\qquad = \mathcal{L}(y_m; t) \qquad (4)$$

$$\text{model variance:} \qquad \text{Variance}(x; h) = \frac{1}{n}\sum_{k=1}^{n}\left(y^{(k)} - y_m\right)^2 \qquad = E_D[\mathcal{L}(h(x); y_m)]. \quad (5)$$

As shown on the above right, these concepts can be expressed entirely in terms of L2 loss $\mathcal{L}$. We can thus extended them to classification (Domingos, 2000) by replacing $\mathcal{L}$ with $\mathcal{L}_{0\text{-}1}$ for classification:

$$\text{0-1 classification loss:} \qquad \mathcal{L}_{0\text{-}1}(y; z) = 0 \text{ if } y = z, \text{ and } 1 \text{ otherwise.} \quad (6)$$

The mean prediction $y_m$ minimizes $\sum_{k=1}^{n}\mathcal{L}_{0\text{-}1}\left(y^{(k)}; y_m\right)$ and becomes the *most often* or *main* prediction. The bias and variance terms become $\mathcal{L}_{0\text{-}1}(y_m; t)$ and $\frac{1}{n}\sum_{k=1}^{n}\mathcal{L}_{0\text{-}1}(y^{(k)}; y_m)$ respectively.

We apply such bias and variance analysis to the CE and long-tail classifiers. We sample CIFAR100 (Krizhevsky, 2009) according to a long-tail distribution multiple times. For each method, we train

a model per long-tail sampled dataset and then estimate the *per-class* bias and variance over these multiple models on the *balanced test set* of CIFAR100-LT Liu et al. (2019). Fig. 1a shows that:

1. **On the model bias**: The head bias is significantly smaller than the tail bias, at $0.3$ vs. $0.9$ for CE. All the existing long-tail methods reduce the overall bias by primarily reducing the tail bias. However, the head-tail bias gap remains large at $0.3$ vs. $0.8$.
2. **On the model variance**: All the existing long-tail methods increase the model variance across all class splits, with a slight reduction in the medium-shot variance for cRT.

That is, existing long-tail methods reduce the model bias for the tail at the cost of increased model variance for all the classes, and the head-tail model bias gap remains large.

We conduct further statistical analysis to understand the head-tail model bias gap. We examine the largest softmax score in the *other* classes of $\{c : c \neq t\}$, where $t$ is the ground-truth class of an instance. The smaller this hardest negative score is, the less the confusion, and the lower the model bias. Fig. 1b shows that there is increasingly more and larger confusion from the head to the tail.

Guided by our model bias/variance and confusion pattern analysis, we propose a new long-tail classifier with four distinctive features: **1)** It reduces the model variance for all the classes with multiple experts. **2)** It reduces the model bias for the tail with an additional distribution-aware diversity loss. **3)** It reduces the computational complexity that comes with multiple experts with a dynamic expert routing module which deploys another trained distinctive expert for a second (or third, ...) opinion only when it is called for. **4)** The routing module and a shared architecture for experts of reduced complexity effectively cut down the computational cost of our multi-expert model, to a level that could be even lower than the commonly adopted baseline with the same backbone.

Our so-called *RoutIng Diverse Experts* (RIDE) not only reduces the model variance for all the classes, but also significantly reduces the model bias for the tail classes and increases the mean accuracies for *all* class splits, all of which existing long-tail methods fail to accomplish.

RIDE delivers 5%∼7% higher accuracies than the current SOTA methods on CIFAR100-LT, ImageNet-LT (Liu et al., 2019) and iNaturalist (Van Horn et al., 2018). RIDE is also a universal framework that can be applied to different backbone networks for improving existing long-tail algorithms such as focal loss (Lin et al., 2017), LDAM (Cao et al., 2019), $\tau$-norm (Kang et al., 2020).

## 2 RELATED WORKS

**Few-shot learning**. To generalize from small training data, meta-learning (Bertinetto et al., 2016; Ravi & Larochelle, 2017; Santoro et al., 2016; Finn et al., 2017; Yang et al., 2018) and data augmentation/generation are two most studied approaches (Chen et al., 2019; Schwartz et al., 2018; Zhang et al., 2019; Liu et al., 2018). Matching Network (Vinyals et al., 2016) and Prototypical Network (Snell et al., 2017) learn discriminative features that can be transferred to new classes through meta-learners without big training data. Hariharan & Girshick (2017), Wang et al. (2018) and Liu et al. (2018) utilize samples from a generative model to augment the training data. However, few-shot learning relies on balanced training data, whereas long-tail recognition has to deal with highly imbalanced training data, e.g., from hundreds in the head to a few instances in the tail.

**Re-balancing/re-weighting**. A direct approach to achieve sample balance is to under- or over-sample training instances according to their class sizes (He & Garcia, 2009). Another option is data augmentation, where additional samples are generated to supplement tail classes, sometimes directly in the feature space (Liu et al., 2020; Chu et al., 2020; Kim et al., 2020). Re-weighting modifies the loss function and puts larger weights on tail classes (Lin et al., 2017; Cui et al., 2019; Cao et al., 2019; Wu et al., 2020) or randomly ignoring gradients from head classes (Tan et al., 2020). However, both sample-wise and loss-wise balancing focus on tail classes, resulting in more sensitivity to fluctuations in the small tail classes and thus much increased model variances (Fig. 1a).

**Knowledge transfer**. OLTR (Liu et al., 2019) and inflated memory (Zhu & Yang, 2020) use memory banks to store and transfer mid- and high-level features from head to tail classes, enhancing feature generalization for the tail. However, this line of work (Liu et al., 2019; Zhu & Yang, 2020; Kang et al., 2020; Jamal et al., 2020; Wang et al., 2017) usually does not have effective control over the knowledge transfer process, often resulting in head performance loss.

**Ensembling and grouping**. One way to counter imbalance is to separate training instances into different groups based their class sizes. Models trained on individual groups are ensembled together in a multi-expert framework. BBN (Zhou et al., 2020) adaptively fuses two-branches that each focus on the head and the tail respectively. LFME (Xiang et al., 2020) distills multiple teacher models into a unified model, each teacher focusing on a relatively balanced group such as many-shot classes, medium-shot classes, and few-shot classes. BBN and LFME still lose head performance and overall generalizability, as no expert has a balanced access to the entire dataset.

**Our RIDE is a non-traditional ensemble method**. **1)** Its experts have shared earlier layers and reduced later channels, less prone to small tail overfitting. **2)** Its experts are jointly optimized. **3)** It deploys experts on an as-needed basis for individual instances with a dynamic expert assignment module. **4)** It reaches higher accuracies with a smaller model complexity and computational cost.

## 3 RIDE: ROUTING DIVERSE DISTRIBUTION-AWARE EXPERTS

We propose a novel multi-expert model (Fig. 2) with shared earlier layers $f_\theta$ and $n$ independent channel-reduced later layers $\Psi = \{\psi_{\theta_1}, ..., \psi_{\theta_n}\}$. They are jointly optimized at Stage 1 and dynamically deployed with a learned expert assignment module at Stage 2. At the inference time, all the $m$ active experts are averaged together in their logits for final ensemble softmax classification:

$$\mathbf{p} = \text{softmax}\left(\frac{1}{m}\sum_{i=1}^{m}\psi_{\theta_i}\left(f_\theta(x)\right)\right). \tag{7}$$

Softmax of the average logits is equivalent to the product of individual classification probabilities, which approximates their joint probability if individual experts makes independent decisions.

**Experts with a shared early backbone and reduced later channels**. Consider $n$ independent experts of the same convolutional neural network (CNN) architecture. Since early layers of a CNN tend to encode generic low-level features, we adopt the common practice in transfer learning and have all the $n$ experts share the same backbone $f_\theta$. Each expert retains independent later layers $\psi_{\theta_i}$, $i = 1, \ldots, n$. To reduce overfitting to small training data in the tail classes, we reduce the number of filter channels in $\psi_{\theta_i}$, e.g., by 1/4. All these $n$ experts are trained together on long-tailed data distribution-aware diversity loss $\mathcal{L}_{\text{D-Diversify}}$ and classification loss $\mathcal{L}_{\text{Classify}}$, such as CE and LDAM.

**Individual expert classification loss**. One way to combine multiple experts is to apply the classification loss to the aggregated logits of individual experts. While this idea works for several recently proposed multi-expert models (Zhou et al., 2020; Xiang et al., 2020), it does not work for our shared experts: Its performance is on-par with an equal-sized single-expert model. Let $\mathcal{L}$ denote the classi-

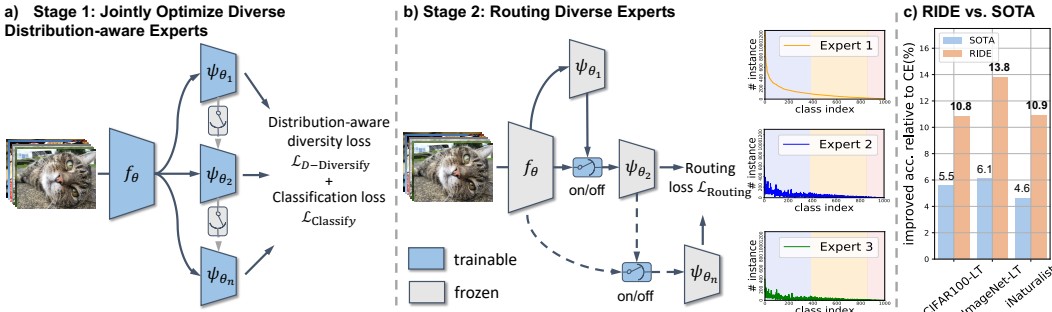

Figure 2: RIDE learns experts and their router in two stages. **a)** We first jointly optimize multiple experts with individual classification losses and mutual distribution-aware diversity losses. **b)** We then train a router that dynamically assigns *ambiguous* samples to additional experts on an as-needed basis. The distribution of instances seen by each expert shows that head instances need fewer experts and the imbalance between classes gets reduced for later experts. At the test time, we collect the logits of assigned experts to make a final decision. **c)** RIDE outperforms SOTA methods (i.e. LFME (Xiang et al., 2020) for CIFAR100-LT, LWS (Kang et al., 2020) for ImageNet-LT and BBN (Zhou et al., 2020) for iNaturalist) on all the benchmarks.

fication loss (e.g. CE) over instance $x$ and its label $y$. We call such an aggregation loss *collaborative*:

$$\mathcal{L}^{\text{collaborative}}(x,y) = \mathcal{L}\left(\frac{1}{n}\sum_{i=1}^{n}\psi_{\theta_i}\left(f_\theta(x)\right), y\right) \quad (8)$$

as it leads to *correlated* instead of *complementary* experts. To discourage correlation, we require each expert to do the job well by itself. Such an aggregation loss is essentially an *individual* loss, and it contributes a large portion of our performance gain in most of our experiments:

$$\mathcal{L}^{\text{individual}}(x,y) = \sum_{i=1}^{n}\mathcal{L}\left(\psi_{\theta_i}\left(f_\theta(x)\right), y\right). \quad (9)$$

**Distribution-aware diversity loss**. The individual classification loss and random initialization lead to diversified experts with a shared backbone. For long-tailed data, we add a regularization term to encourage complementary decisions from multiple experts. That is, we maximize the KL-divergence between different experts' classification probabilities on instance $x$ in class $y$ over a total of $c$ classes:

diversity loss: $\qquad \mathcal{L}_{\text{D-Diversify}}(x,y;\theta_i) = \dfrac{-1}{n-1}\sum_{j=1,j\neq i}^{n} D_{\text{KL}}\left(\mathbf{p}^{(i)}(x,y)\|\mathbf{p}^{(j)}(x,y)\right) \quad (10)$

KL divergence: $\qquad D_{\text{KL}}(\mathbf{p}\|\mathbf{q}) = \sum_{k=1}^{c}\mathbf{p}_k\log\left(\dfrac{\mathbf{p}_k}{\mathbf{q}_k}\right) \quad (11)$

classification by $\theta_i$: $\qquad \mathbf{p}^{(i)}(x,y) = \text{softmax}\left(\left[\dfrac{\psi_{\theta_i}(f_\theta(x))_1}{T_1} \quad \dots \quad \dfrac{\psi_{\theta_i}(f_\theta(x))_c}{T_c}\right]\right). \quad (12)$

We vary the temperature $T$ (or *concentration*) (Hadsell et al., 2006; Wu et al., 2018) applied to class $k$'s logit $\psi_{\theta_i}(f_\theta(x))_k$: For class $k$ with $n_k$ instances, the smaller the $n_k$, the lower the temperature $T_k$, the more sensitive the classification probability $\mathbf{p}$ is to a change in the feature $\psi$. Specifically,

class-wise temperature: $\qquad T_k = \alpha\left(\beta_k + 1 - \max_j \beta_j\right) \quad (13)$

normalized class size: $\qquad \beta_k = \gamma \cdot \dfrac{n_k}{\frac{1}{c}\sum_{s=1}^{c}n_s} + (1-\gamma). \quad (14)$

$T$ scales linearly with the class size, ensuring $\beta_k = 1, T_k = \alpha$ for a balanced set. This simple adaptation allows us to find classifiers of enough complexity for the head and enough robustness for the tail: On one hand, we need strong classifiers to handle large sample variations within head classes; on the other hand, such classifiers are prone to overfit small training data within tail classes. We adapt the temperature only after the CNN network is trained for several epochs and the feature is stabilized, similar to the training scheme for deferred reweighting (Cao et al., 2019).

**Joint expert optimization.** For our $n$ experts $\theta_1, \dots, \theta_n$ with a shared backbone $\theta$, we optimize their individual classification losses ($\mathcal{L}_{\text{Classify}} = \mathcal{L}$, any classification loss such as CE, LDAM, and focal loss) and their mutual distribution-aware divergency losses, weighted by hyperparameter $\lambda$:

$$\mathcal{L}_{\text{Total}}(x,y) = \sum_{i=1}^{n}\left(\mathcal{L}_{\text{Classify}}(x,y;\theta_i) + \lambda \cdot \mathcal{L}_{\text{D-Diversify}}(x,y;\theta_i)\right). \quad (15)$$

Since these loss terms are completely symmetrical with respect to each other, the $n$ experts learned at Stage 1 are equally good and distinctive from each other.

**Routing diversified experts**. To cut down the test-time computational cost that comes with multiple experts, we train a router at Stage 2 to deploy these (arbitrarily ordered) experts sequentially on an as-needed basis. Assume that the first $k$ experts have been deployed for instance $x$. The router takes in the image feature and the mean logits from the first to the $k$-th expert, and makes a binary decision $y_{\text{on}}$ on whether to deploy the $k+1$-th expert. If the $k$-th expert wrongly classifies $x$, but one of the rest $n-k$ experts correctly classifies $x$, ideally the router should switch on, i.e., output $y_{\text{on}} = 1$, and otherwise $y_{\text{on}} = 0$. We construct a simple binary classifier with two fully connected layers to learn each router. Each of the $n-1$ routers for $n$ experts has a shared component to reduce the feature dimensions and an individual component to make decisions.

Specifically, we normalize the image feature $f_\theta(x)$ (for training stability), reduce the feature dimension (to e.g. 16 in our experiments) by a fully connected layer $\mathbf{W}_1$ which is shared with all routers, followed by ReLU and flattening, concatenate with the top-$s$ ranked mean logits from the first to $k$-th expert $\frac{1}{k}\sum_{i=1}^{k}\psi_{\theta_i}(f_\theta(x))$, project it to a scalar by $\mathbf{W}_2^{(k)}$ which is independent between routers, and finally apply Sigmoid function $S(x) = \frac{1}{1+e^{-x}}$ to get a continuous activation value in [0,1]:

$$\text{router activation: } r(x) = S\left(\mathbf{W}_2^{(k)}\begin{bmatrix} \text{flatten} \cdot \text{ReLU}\left(\mathbf{W}_1\frac{f_\theta(x)}{\|f_\theta(x)\|}\right) \\ \frac{1}{k}\sum_{i=1}^{k}\psi_{\theta_k}(f_\theta(x))|_{\text{top-}s\text{-components}} \end{bmatrix}\right). \tag{16}$$

The router has a negligible size and compute, where $s$ ranges from 30 for CIFAR100 to 50 for iNaturalist (8,142 classes). It is optimized with a weighted variant of binary CE loss:

$$\mathcal{L}_{\text{Routing}}(r(x), y_{\text{on}}) = -\omega_{\text{on}}\, y_{\text{on}} \log\left(r(x)\right) - (1 - y_{\text{on}}) \log\left(1 - r(x)\right) \tag{17}$$

where $\omega_{\text{on}}$ controls the easiness to switch on the router. We find $\omega_{\text{on}} = 100$ to be a good trade-off between classification accuracy and computational cost for all our experiments. At the test time, we simply threshold the activation with 0.5: If $r(x) < 0.5$, the classifier makes the final decision with the current collective logits, otherwise it proceeds to the next expert.

**Optional self-distillation**. While existing long-tail classifiers such as BBN Zhou et al. (2020) and LFME Xiang et al. (2020) have a fixed number of experts, our method could have an arbitrary number of experts to balance classification accuracy and computation cost. We can optionally apply self-distillation from a model with more (6 in our setting) experts to the same model with fewer experts for further performance gain (0.4%~0.8% for most experiments). We choose knowledge distillation (Hinton et al., 2015) by default. Implementation details and comparisons with various distillation algorithms such as CRD (Tian et al., 2019) are investigated in Appendix Section A.2.

## 4 EXPERIMENTS

We experiment on major long-tailed recognition benchmarks and various backbone networks.

1. **CIFAR100-LT** (Cao et al., 2019): CIFAR100 is sampled by class per an exponential decay across classes. We choose imbalance factor 100 and ResNet-32 (He et al., 2016) backbone.

2. **ImageNet-LT** (Liu et al., 2019): Multiple backbone networks are experimented on ImageNet-LT, including ResNet-10, ResNet-50 and ResNeXt-50 (Xie et al., 2017). All backbone networks are trained with a batch size of 256 on 8 RTX 2080Ti GPUs for 100 epochs using SGD with an initial learning rate of 0.1 decayed by 0.1 at 60 epochs and 80 epochs. See more details and results on other backbones in Appendix.

3. **iNaturalist 2018** (Van Horn et al., 2018): It is a naturally imbalanced fine-grained dataset with 8,142 categories. We use ResNet-50 as the backbone and apply the same training recipe as for ImageNet-LT except batch size 512, as in (Kang et al., 2020).

**CIFAR100-LT Results**. Table 1 shows that RIDE outperforms SOTA by a large margin on CIFAR100-LT. The average computational cost is even about 10% less than baseline models with two experts as in BBN. RIDE surpasses multi-expert methods, LFME (Xiang et al., 2020) and BBN (Zhou et al., 2020), by more than 5.3% and 6.5% respectively.

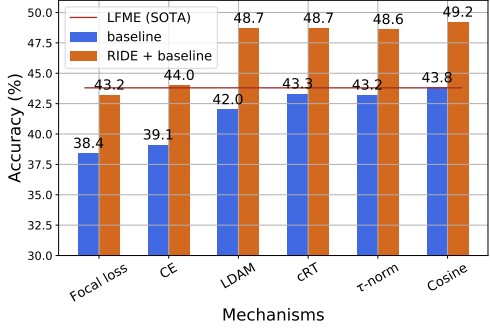

Figure 3: **RIDE is a universal framework** that can be extended to various long-tail recognition methods and obtain a consistent top-1 accuracy increase. RIDE is experimented on CIFAR100-LT and applied to various training mechanisms. By using RIDE, cross-entropy loss (*without any re-balancing strategies*) can even outperforms previous SOTA method on CIFAR100-LT. Although higher accuracy can be obtained using distillation, we did not apply it here.

Table 1: RIDE achieves the state-of-the-art results on **CIFAR100-LT** *without* sacrificing the performance of many-shot classes like all previous methods. Compared with BBN (Zhou et al., 2020) and LFME (Xiang et al., 2020), which also contain multiple experts (or branches), RIDE (2 experts) outperforms them by a large margin with fewer GFlops. The relative computation cost (averaged on testing set) with respect to the baseline model and absolute improvements against SOTA (colored in green) are reported. † denotes our reproduced results with released code. ‡ denotes results copied from (Cao et al., 2019) and the imbalance ratio is 100.

| Methods | MFlops | All | Many | Med | Few |
|---|---|---|---|---|---|
| Cross Entropy (CE) ‡ | 69.5 (1.0x) | 38.3 | - | - | - |
| Cross Entropy (CE) † | 69.5 (1.0x) | 39.1 | 66.1 | 37.3 | 10.6 |
| Focal Loss ‡ (Lin et al., 2017) | 69.5 (1.0x) | 38.4 | - | - | - |
| OLTR † (Liu et al., 2019) | - | 41.2 | 61.8 | 41.4 | 17.6 |
| LDAM + DRW (Cao et al., 2019) | 69.5 (1.0x) | 42.0 | - | - | - |
| LDAM + DRW † (Cao et al., 2019) | 69.5 (1.0x) | 42.0 | 61.5 | 41.7 | 20.2 |
| BBN (Zhou et al., 2020) | 74.3 (1.1x) | 42.6 | - | - | - |
| $\tau$-norm † (Kang et al., 2020) | 69.5 (1.0x) | 43.2 | 65.7 | 43.6 | 17.3 |
| cRT † (Kang et al., 2020) | 69.5 (1.0x) | 43.3 | 64.0 | 44.8 | 18.1 |
| M2m (Kim et al., 2020) | - | 43.5 | - | - | - |
| LFME (Xiang et al., 2020) | - | 43.8 | - | - | - |
| RIDE (2 experts) | **64.8 (0.9x)** | 47.0 (+3.2) | 67.9 | 48.4 | 21.8 |
| RIDE (3 experts) | 77.8 (1.1x) | 48.0 (+4.2) | 68.1 | 49.2 | 23.9 |
| RIDE (4 experts) | 91.9 (1.3x) | **49.1 (+5.3)** | **69.3** | **49.3** | **26.0** |

Table 2: RIDE achieves state-of-the-art results on **ImageNet-LT** (Liu et al., 2019) and obtains consistent performance improvements on various backbones. The top-1 accuracy and computational cost are compared with the state-of-the-art methods on ImageNet-LT, with ResNet-50 and ResNeXt-50 as the backbone networks. Results marked with † are copied from (Kang et al., 2020). Detailed results on each split are listed in appendix materials.

| Methods | ResNet-50 | | ResNeXt-50 | |
|---|---|---|---|---|
| | GFlops | Acc. (%) | GFlops | Acc. (%) |
| Cross Entropy (CE) † | 4.11 (1.0x) | 41.6 | 4.26 (1.0x) | 44.4 |
| OLTR † (Liu et al., 2019) | - | - | - | 46.3 |
| NCM (Kang et al., 2020) | 4.11 (1.0x) | 44.3 | 4.26 (1.0x) | 47.3 |
| $\tau$-norm (Kang et al., 2020) | 4.11 (1.0x) | 46.7 | 4.26 (1.0x) | 49.4 |
| cRT (Kang et al., 2020) | 4.11 (1.0x) | 47.3 | 4.26 (1.0x) | 49.6 |
| LWS (Kang et al., 2020) | 4.11 (1.0x) | 47.7 | 4.26 (1.0x) | 49.9 |
| RIDE (2 experts) | **3.71 (0.9x)** | 54.4 (+6.7) | **3.92 (0.9x)** | 55.9 (+6.0) |
| RIDE (3 experts) | 4.36 (1.1x) | 54.9 (+7.2) | 4.69 (1.1x) | 56.4 (+6.5) |
| RIDE (4 experts) | 5.15 (1.3x) | **55.4 (+7.7)** | 5.19 (1.2x) | **56.8 (+6.9)** |

**RIDE as a universal framework**. Fig. 3 shows that RIDE consistently benefits from better loss functions and training processes. Whether the model is trained end-to-end (*focal loss, CE, LDAM*) or in two stages (*cRT, $\tau$-norm, cosine*), RIDE delivers consistent accuracy gains. In particular, RIDE with a simple cosine classifier, which we constructed by normalizing the classifier weights and retraining them with a long-tail re-sampling strategy (similar to cRT), achieves on-par performance with the current SOTA methods. Fig. 3 also shows that two-stage methods are generally better than single-stage ones. Nevertheless, since they require an additional training stage, for simplicity, we use the single-stage LDAM as the default $\mathcal{L}_{\text{Classify}}$ in RIDE throughout our remaining experiments.

**ImageNet-LT Results**. Table 2 shows that RIDE outperforms SOTA, LWS and cRT, by more than 7.7% with ResNet-50. ResNeXt-50 is based on group convolution (Xie et al., 2017), which divides all filters into several groups and aggregates information from multiple groups. ResNeXt-50 generally performs better than ResNet-50 on multiple tasks. It provides 6.9% gain on ImageNet-LT.

**iNaturalist 2018 Results**. Table 3 shows that RIDE outperforms current SOTA by 6.3%. Surprisingly, RIDE obtains very similar results on many-shots, medium-shots and few-shots, ideal for long tailed recognition. Current SOTA method BBN also uses multiple experts; however, it significantly

Table 3: RIDE outperforms previous state-of-the-art methods on challenging **iNaturalist 2018** (Van Horn et al., 2018) dataset, which contains 8,142 classes, by a large margin. Relative improvements to SOTA result of each split (colored with gray) are also listed, with the largest boost from few-shot classes. Compared with previous SOTA method BBN, which also contains multiple "experts", RIDE achieves more than 20% higher top-1 accuracy on many-shot classes. Results marked with † are from BBN (Zhou et al., 2020) and Decouple (Kang et al., 2020). BBN's results are from the released checkpoint.

| Methods | GFlops | All | Many | Medium | Few |
|---------|--------|-----|------|--------|-----|
| CE † | 4.14 (1.0x) | 61.7 | 72.2 | 63.0 | 57.2 |
| CB-Focal † | 4.14 (1.0x) | 61.1 | - | - | - |
| OLTR | 4.14 (1.0x) | 63.9 | 59.0 | 64.1 | 64.9 |
| LDAM + DRW † | 4.14 (1.0x) | 64.6 | - | - | - |
| cRT | 4.14 (1.0x) | 65.2 | 69.0 | 66.0 | 63.2 |
| $\tau$-norm | 4.14 (1.0x) | 65.6 | 65.6 | 65.3 | 65.9 |
| LWS | 4.14 (1.0x) | 65.9 | 65.0 | 66.3 | 65.5 |
| BBN | 4.36 (1.1x) | 66.3 | 49.4 | 70.8 | 65.3 |
| RIDE (2 experts) | **3.67 (0.9x)** | 71.4 (+5.1) | 70.2 (+1.2) | 71.3 (+0.5) | 71.7 (+5.8) |
| RIDE (3 experts) | 4.17 (1.0x) | 72.2 (+5.9) | 70.2 (+1.2) | 72.2 (+1.4) | 72.7 (+6.8) |
| RIDE (4 experts) | 4.51 (1.1x) | **72.6 (+6.3)** | **70.9 (+1.9)** | **72.4 (+1.6)** | **73.1 (+7.2)** |

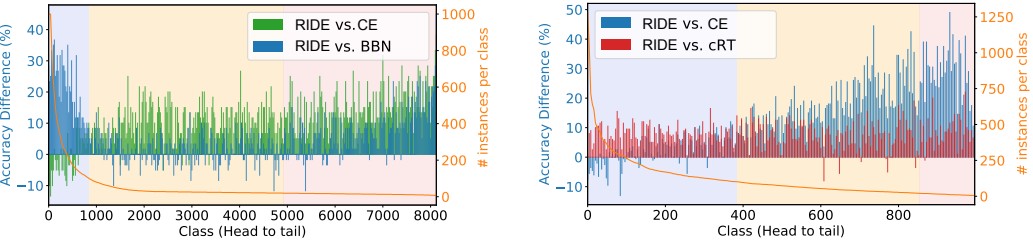

Figure 4: Compared to SOTAs, RIDE improves top-1 accuracy on all three splits (many-/med-/few-shot). The absolute accuracy differences of RIDE (blue) over *iNaturalist*'s current state-of-the-art method BBN (Zhou et al., 2020) (**left**) and *ImageNet-LT*'s current state-of-the-art method cRT (Kang et al., 2020) (**right**) are shown. RIDE improves the performance of few- and medium-shots categories without sacrificing the accuracy on many-shots, and outperforms BBN on many-shots by a large margin.

decreases the performance on many-shots by about 23%. RIDE is remarkable at increasing the few-shot accuracy without reducing the many-shot accuracy.

**Comparing with SOTAs on iNaturalist and ImageNet-LT**. As illustrated in Fig. 4, our approach provides a comprehensive treatment to all the many-shot, medium-shot and few-shot classes, achieving substantial improvements to current state-of-the-art on all aspects. Compared with cRT which reduces the performance on the many-shot classes, RIDE can achieves significantly better performance on the few-shot classes without impairing the many-shot classes. Similar observations can be obtained in the comparison with the state-of-the-art method BBN (Zhou et al., 2020) on iNaturalist.

**Contribution of each component of RIDE.** RIDE is jointly trained with $\mathcal{L}_{\text{D-Diversify}}$ and $\mathcal{L}_{\text{Classify}}$, we use LDAM for $\mathcal{L}_{\text{Classify}}$ by default. Table 4 shows that the architectural change from the original ResNet-32 to the RIDE variant with $2 \sim 4$ experts contributes $2.7\% \sim 4.3\%$ gain. Applying the individual classification loss instead of the collaborative loss brings 1.5% gain. Adding the diversity loss further improves about 0.9%. The computational cost is greatly reduced by adding the dynamic expert router. Knowledge distillation from RIDE with 6 experts obtains another 0.6% gain. All these components deliver 7.1% gain over baseline LDAM.

**Impact of the number of experts.** Fig. 5 shows that whether in terms of relative or absolute gains, few-shots benefit more with more experts. For example, the relative gain is 16% vs. 3.8% for few-shots and many-shots respectively. No distillation is applied in this comparison.

Table 4: **Ablation studies** on the effectiveness of each component on CIFAR100-LT. LDAM is used as our classification loss. The first 3 RIDE models only have architectural change without changes in training method. The performance without $\mathcal{L}_{\text{Individual}}$ checked indicates directly applying classification loss onto the final model output, which is the mean expert logits. This is referred to as collaborative loss above. In contrast, if $\mathcal{L}_{\text{Individual}}$ if checked, we apply individual loss to each individual expert. The difference between collaborative loss and individual loss is described above. By adding the router module, the computational cost of RIDE can be significantly reduced, while the accuracy degradation is negligible. Knowledge distillation step is optional if further improvements are desired. Various knowledge distillation techniques are compared in the appendix.

| Methods | #expert | $\mathcal{L}_{\text{Individual}}$ | $\mathcal{L}_{\text{D-Diversify}}$ | Router | distill | GFlops | Acc. (%) |
|---|---|---|---|---|---|---|---|
| LDAM + DRW | 1 | | | | | | 42.0 |
| RIDE | 2 | | | | | 1.1x | 44.7 **(+2.7)** |
| | 3 | | | | | 1.5x | 46.1 **(+4.1)** |
| | 4 | | | | | 1.8x | 46.3 **(+4.3)** |
| | 4 | ✓ | | | | 1.8x | 47.8 **(+5.8)** |
| | 4 | ✓ | ✓ | | | 1.8x | 48.7 **(+6.7)** |
| | 4 | ✓ | ✓ | | ✓ | 1.8x | **49.3** **(+7.3)** |
| | 4 | ✓ | ✓ | ✓ | ✓ | 1.3x | 49.1 **(+7.1)** |

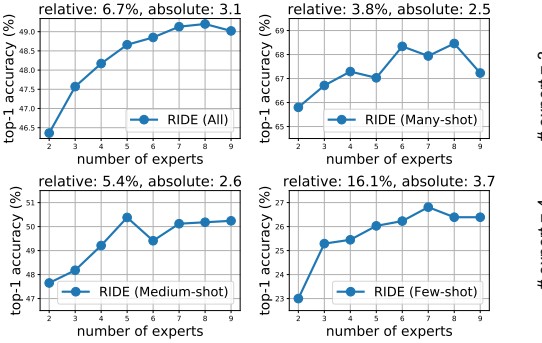

Figure 5: **# experts vs. top-1 accuracy for each split** (All, Many/Medium/Few) of CIFAR100-LT. Compared with the many-shot split, which is 3.8% relatively improved by adding more experts, the few-shot split can get more benefits, that is, a relative improvement of 16.1%.

Figure 6: **The proportion of the number of experts** allocated to each split of CIFAR100-LT. For RIDE with 3 or 4 experts, more than half of many-shot instances only require one expert. On the contrary, more than 76% samples of few-shot classes require opinions from additional experts.

**The number of experts allocated to each split.** Fig. 6 shows that instances in few-shots need more experts whereas most instances in many-shots just need the first expert. That is, low confidence in tail instances often requires the model to seek a second (or a third, ...) opinion.

## 5 SUMMARY

We take a dynamic view of training data and study long-tailed recognition with model bias and variance analysis. Existing long-tail classifiers do not reduce head-tail model bias gap enough while increasing model variance across all the classes. We propose a novel multi-expert model called RIDE to reduce model biases and variances throughout. It trains partially shared diverse distribution-aware experts and routes an instance to additional experts when necessary, with computational costs comparable to a single expert. RIDE outperforms SOTA by a large margin. It is also a universal framework that works with various backbones and training schemes for consistent gains.

**Acknowledgments.** This work was supported, in part, by Berkeley Deep Drive, US Government fund through Etegent Technologies on Low-Shot Detection and Semi-supervised Detection, and NTU NAP and A*STAR via Industry Alignment Fund: Industry Collaboration Projects Grant.

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

# A  APPENDIX

## A.1  DATASETS AND IMPLEMENTATIONS

We conduct experiments on three major long-tailed recognition benchmarks and different backbone networks to prove the effectiveness and universality of RIDE:

1.**CIFAR100-LT** (Krizhevsky, 2009; Cao et al., 2019): The original version of CIFAR-100 contains 50,000 images on training set and 10,000 images on validation set with 100 categories. The long-tailed version of CIFAR-100 follows an exponential decay in sample sizes across different categories. We conduct experiment on CIFAR100-LT with an imbalance factor of 100, i.e. the ratio between the most frequent class and the least frequent class.

To make fair comparison with previous works, we follow the training recipe of (Cao et al., 2019) on CIFAR100-LT. We train the ResNet-32 (He et al., 2016) backbone network by SGD optimizer with a momentum of 0.9. CIFAR100-LT is trained for 200 epochs with standard data augmentations (He et al., 2016) and a batch size of 128 on one RTX 2080Ti GPU. The learning rate is initialized as 0.1 and decayed by 0.01 at epoch 120 and 160 respectively.

2.**ImageNet-LT** (Deng et al., 2009; Liu et al., 2019): ImageNet-LT is constructed by sampling a subset of ImageNet-2012 following the Pareto distribution with the power value $\alpha = 6$ (Liu et al., 2019). ImageNet-LT consists of 115.8k images from 1,000 categories, with the largest and smallest categories containing 1,280 and 5 images, respectively.

Multiple backbone networks are experimented on ImageNet-LT, including ResNet-10, ResNet-50 and ResNeXt-50 (Xie et al., 2017). All backbone networks are trained with a batch size of 256 on 8 RTX 2080Ti GPUs for 100 epochs using SGD with an initial learning rate of 0.1 decayed by 0.1 at 60 epochs and 80 epochs. We utilize standard data augmentations as in (He et al., 2016).

3.**iNaturalist** (Van Horn et al., 2018): The iNaturalist-2018 dataset is an imbalanced datasets with 437,513 training images from 8,142 classes with a balanced test set of 24,426 images. We use ResNet-50 as the backbone network and apply the same training recipe as ImageNet-LT, except that we use a batch size of 512.

## A.2  ADDITIONAL EXPERIMENTS

**Ablation study on distillation methods.** Self-distillation step is optional but recommended if further improvements (0.4%∼0.8% for most experiments) are desired. We apply distillation from a more powerful model with more experts into a model with fewer experts. A simple way to transfer knowledge is knowledge distillation (KD) (Hinton et al., 2015), which applies KD loss ($\mathcal{L}_{\mathrm{KD}} = T^2 D_{\mathrm{KL}}(\vec{l}_{\mathrm{teacher}}/T, \vec{l}_{\mathrm{expert}_i}/T)$) to match the distribution of logits of a teacher and a student. We found that for teacher model with more experts using smaller distillation loss factor gives better performance. We hypothesize that since we distill from the same teachers, giving large distillation factor prevents the branches from becoming as diversified as it is able to. We also explored other distillation methods, such as CRD (Tian et al., 2019), PKT (Passalis & Tefas, 2018), and SP (Tung & Mori, 2019), and compared the differences in Table 5. Although adding other methods along with KD may boost performance, the difference is small. Therefore, we opt for simplicity and use KD only unless otherwise noticed.

**Detailed results for ImageNet-LT experiments**. We list details of our ResNet-50 experiments in ImageNet-LT on Table 7. With 2 experts, we are able to achieve about 7% gain in accuracy with computational cost about 10% less than baseline. In contrast to previous methods that sacrifice many-shot accuracy to get few-shot accuracy, we improve on all three splits on ImageNet-LT. From 3 experts to 4 experts, we keep the same many-shot accuracy while increasing the few-shot accuracy, indicating that we are using the additional computational power to improve on the hardest part of the data rather than uniformly applying to all samples.

We also list our ResNet-10 and ResNeXt-50 experiments on Table 6 and 8, respectively, to compare against other works evaluated on these backbones. Our method also achieves lower computational cost and higher performance when compared to other methods.

**Comparison with ensemble method.** Since our method requires the joint decision from several experts, which raw ensembles also do, we also compare against ensembles of LDAM in Fig.7 on

Table 5: **Comparison of different distillation methods.** We transfer from a model based on ResNet-32 with 6 experts to a model of the same type, except with fewer experts. We use CIFAR100-LT for the following comparison. No expert assignment module is used in the following experiments. Following the procedure for CRD (Tian et al., 2019), we also apply KD when we transfer from a teacher to students with other distillation methods.

| Model Type | #expert | Distillation Method | Accuracy (%) |
|---|---|---|---|
| Teacher | 6 | | 49.7 |
| Student | 2 | No Distillation | 46.6 |
| | 2 | KD (Hinton et al., 2015) | 47.3 |
| | 2 | CRD (Tian et al., 2019) | **47.5** |
| | 2 | PKT (Passalis & Tefas, 2018) | 47.2 |
| | 2 | SP (Tung & Mori, 2019) | 47.2 |
| | 3 | No Distillation | 47.9 |
| | 3 | KD (Hinton et al., 2015) | 48.4 |
| | 3 | CRD (Tian et al., 2019) | 48.5 |
| | 3 | PKT (Passalis & Tefas, 2018) | 48.3 |
| | 3 | SP (Tung & Mori, 2019) | **48.7** |
| | 4 | No Distillation | 48.7 |
| | 4 | KD (Hinton et al., 2015) | **49.3** |
| | 4 | CRD (Tian et al., 2019) | 49.0 |
| | 4 | PKT (Passalis & Tefas, 2018) | 48.9 |
| | 4 | SP (Tung & Mori, 2019) | 49.0 |

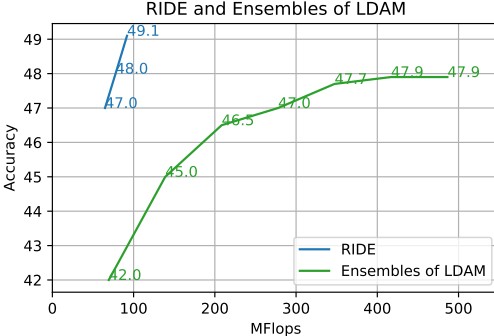

Figure 7: Comparison between our method and multiple LDAM models ensembled together. In the figure, ensembles of LDAM start from 1 ensemble (original LDAM) to 7 ensembles, and RIDE starts from 2 experts to 4 experts. Our method achieves higher accuracy with substantially less computational cost compared to ensemble method.

CIFAR100-LT. In the figure, even our method with 4 experts has less computational cost than the minimum computational cost for the ensemble of 2 LDAM models. This indicates that our model is much more efficient and powerful in terms of computational cost and accuracy than ensemble on long-tailed datasets.

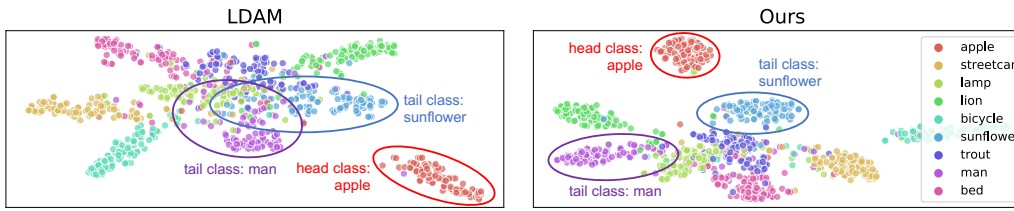

Figure 8: t-SNE visualization of LDAM's and our model's embedding space of CIFAR100-LT. The feature embedding of RIDE is more compact for both head and tail classes and better separated. This behavior greatly reduces the difficulty for the classifier to distinguish the tail category.

**t-SNE visualization**. We also provide the t-SNE visualization of embedding space on CIFAR100-LT as in Fig. 8. Compared with the baseline method LDAM, the feature embedding of RIDE is more compact for both the head and tail classes and better separated from the neighboring classes. This greatly reduces the difficulty for the classifier to distinguish the tail category.

**What if we apply RIDE to balanced datasets?** We also conducted experiments on CIFAR100 to check if our method can achieve similar performance gains on balanced datasets. However, we only obtained an improvement of about 1%, which is much smaller than the improvements observed on the CIFAR100-LT. Compared with balanced datasets, long-tailed datasets can get more benefits from RIDE.

Table 6: Top-1 accuracy comparison with state-of-the-art methods on **ImageNet-LT** (Liu et al., 2019) with **ResNet-10**. Performance on Many-shot (>100), Medum-shot (≤100 & >20) and Few-shot (≤20) are also provided. Results marked with † are copied from (Liu et al., 2019). Results with ‡ are from (Xiang et al., 2020).

| Methods | GFlops | Many | Medium | Few | Overall |
|---|---|---|---|---|---|
| Cross Entropy (CE) † | 0.89 (1.0x) | 40.9 | 10.7 | 0.4 | 20.9 |
| Focal Loss † (Lin et al., 2017) | 0.89 (1.0x) | 36.4 | 29.9 | 16.0 | 30.5 |
| Range Loss † (Zhang et al., 2017) | 0.89 (1.0x) | 35.8 | 30.3 | 17.6 | 30.7 |
| Lifted Loss † (Oh Song et al., 2016) | 0.89 (1.0x) | 35.8 | 30.4 | 17.9 | 30.8 |
| OLTR (Liu et al., 2019) | 0.89 (1.0x) | 43.2 | 35.1 | 18.5 | 35.6 |
| LFME (Xiang et al., 2020) | - | 47.0 | 37.9 | 19.2 | 38.8 |
| Many-shot only ‡ | - | 59.3 | | | |
| Medium-shot only ‡ | - | | 35.9 | | |
| Few-shot only ‡ | - | | | 14.3 | |
| RIDE (2 experts) | **0.85 (1.0x)** | 57.5 | 40.8 | 26.9 | 45.3 (+6.5) |
| RIDE (3 experts) | 0.97 (1.1x) | 57.6 | 41.7 | **28.0** | 45.9 (+7.1) |
| RIDE (4 experts) | 1.07 (1.2x) | **58.5** | **42.4** | 27.7 | **46.6** (+7.8) |

Table 7: Top-1 accuracy comparison with state-of-the-art methods on **ImageNet-LT** (Liu et al., 2019) with **ResNet-50**. Performance on Many-shot (>100), Medum-shot (≤100 & >20) and Few-shot (≤20) are also provided. Results marked with † are copied from (Kang et al., 2020).

| Methods | GFlops | Many | Medium | Few | Overall |
|---|---|---|---|---|---|
| Cross Entropy (CE) † | 4.11 (1.0x) | 64.0 | 33.8 | 5.8 | 41.6 |
| NCM (Kang et al., 2020) | - | 53.1 | 42.3 | 26.5 | 44.3 |
| cRT (Kang et al., 2020) | 4.11 (1.0x) | 58.8 | 44.0 | 26.1 | 47.3 |
| $\tau$-norm (Kang et al., 2020) | 4.11 (1.0x) | 56.6 | 44.2 | 27.4 | 46.7 |
| LWS (Kang et al., 2020) | 4.11 (1.0x) | 57.1 | 45.2 | 29.3 | 47.7 |
| RIDE (2 experts) | **3.71 (0.9x)** | 65.8 | 51.0 | 34.6 | 54.4 (+6.7) |
| RIDE (3 experts) | 4.36 (1.1x) | **66.2** | 51.7 | 34.9 | 54.9 (+7.2) |
| RIDE (4 experts) | 5.15 (1.3x) | **66.2** | **52.3** | **36.5** | **55.4** (+7.7) |

Table 8: Top-1 accuracy comparison with state-of-the-art methods on **ImageNet-LT** (Liu et al., 2019) with **ResNeXt-50**. Performance on Many-shot (>100), Medum-shot (≤100 & >20) and Few-shot (≤20) are also provided. Results marked with † are copied from (Kang et al., 2020).

| Methods | GFlops | Many | Medium | Few | Overall |
|---|---|---|---|---|---|
| Cross Entropy (CE) † | 4.26 (1.0x) | 65.9 | 37.5 | 7.7 | 44.4 |
| NCM (Kang et al., 2020) | - | 56.6 | 45.3 | 28.1 | 47.3 |
| cRT (Kang et al., 2020) | 4.26 (1.0x) | 61.8 | 46.2 | 27.4 | 49.6 |
| $\tau$-norm (Kang et al., 2020) | 4.26 (1.0x) | 59.1 | 46.9 | 30.7 | 49.4 |
| LWS (Kang et al., 2020) | 4.26 (1.0x) | 60.2 | 47.2 | 30.3 | 49.9 |
| RIDE (2 experts) | **3.92 (0.9x)** | 67.6 | 52.5 | 35.0 | 55.9 (+6.0) |
| RIDE (3 experts) | 4.69 (1.1x) | 67.6 | 53.5 | 35.9 | 56.4 (+6.5) |
| RIDE (4 experts) | 5.19 (1.2x) | **68.2** | **53.8** | **36.0** | **56.8** (+6.9) |

