# OpenReview forum: "Long-tailed Recognition by Routing Diverse Distribution-Aware Experts"
_ICLR.cc/2021/Conference — ICLR 2021 Spotlight_

### Official Review · AnonReviewer2 · 2020-10-25
**A good paper that shows a new direction for long-tailed classification.**

**Rating:** 7
**Confidence:** 4

**Review:**

##########################################################################

Summary:

This paper proposes a Routing Diverse Experts (RIDE) framework to solve the long-tailed classification problem. It has 1) a shared low-level feature extractor and multiple expert classifiers, 2) a distribution-aware diversity loss to encourage experts learning different classification strategies, 3) an expert routing module that dynamically selects a subset of experts for each test instance to make a joint decision. This paper firstly increases the performances on all three splits (many-/med-/few-shot), while most of the existing methods have to sacrifice the head for tail improvements.

##########################################################################

Pros:

+ The proposed Bias & Variance metrics provide a better understanding of why previous work almost always improves the tail accuracy but decreases head performance(trade-off between bias and var).
+ To the best of my knowledge, this paper firstly increases the performances on both head and tail.
+ The proposed RIDE framework can be applied to a variety of long-tailed recognition methods.
+ Although RIDE includes multiple experts, each expert only has 1/4 channels, which allows overall computation overhead even decreases when using 2 experts.

##########################################################################

Concerns:

I'm overall satisfied with this paper, but I do have a few concerns:
- The proposed RIDE does look like an ensemble method to me. Although you provide a comparison with the ensemble method in the appendix, I noticed that 2 ensembles will achieve 45.0 accuracies on CIFAR100-LT, which is exactly the same as 2 experts + L_diversity in Table 4. Although RIDE can achieve better performance when the number of ensemble models/experts is increased to 3 or 4, does it means that diversity loss only works on a larger number of experts?

- The expert routing/assignment module dynamically routes cascaded experts, and all the passed experts before the stop will jointly make a decision. How to make sure the early experts are the more robust ones? Since experts have already been frozen when the routing module is trained, and the diversity loss won't force the early experts to be more robust. There is no guarantee that the involved early experts are more useful than the skipped ones.

##########################################################################

Questions during the rebuttal period:

Please address and clarify the concerns above. Besides, I think the experts in RIDE may learn to be biased toward different categories due to the diversity loss, and the routing module works as an outlier detector which only stops when it finds a proper expert biased toward the underline ground-truth labels. Is it another way to understand RIDE? or do I misunderstand some parts of our paper?

##########################################################################

Reasons for score:

Overall, I vote for accepting. I think this paper shows a new direction for long-tailed classification that we don't need to sacrifice the head performance for a balanced classifier anymore.

---

> ### Author Response · Authors · 2020-11-23
> **Thanks so much for your constructive and encouraging feedback! (Part 1/3)**
>
> Thanks so much for your constructive and encouraging feedback! Below we address your concerns.
>
> **[Q1]** I noticed that 2 ensembles will achieve 45.0 accuracies on CIFAR100-LT, which is exactly the same as 2 experts + $L_{\text{Diversity}}$ in Table 4. Although RIDE can achieve better performance when the number of ensemble models/experts is increased to 3 or 4, does it means that diversity loss only works on a larger number of experts?
>
> **[A1]** Sorry about the confusion!  To better understand the effectiveness of our distribution-aware diversity loss $L_{\text{D-Diversity}}$, we conducted experiments on RIDE with $L_{\text{Diversity}}$ and $L_{\text{D-Diversity}}$ losses respectively. $L_{\text{Diversity}}$  denotes the diversity loss with the same temperature for all classes and $L_{\text{D-Diversity}}$ denotes the diversity loss with a distribution-aware temperature for each class. $L_{\text{D-Diversity}}$ is our final proposed method and is used in most of the experiments. See Section 3.2 of our paper for details.
>
> Here is a quick summary of our experiments with baseline method (LDAM DRW), an ensemble of 2 baseline models, and our method with 2 experts. Table 4 shows that RIDE with $L_{\text{Diversity}}$ and $L_{\text{D-Diversity}}$ loss achieves a top-1 accuracy of 45.0% and 46.6% respectively. RIDE with $L_{\text{D-Diversity}}$ loss (2 experts) outperforms an ensemble of two LDAM models by more than 1.6% with only a half of the compute. These results demonstrate that the distribution-aware diversity loss is both effective and efficient even with a small number of experts.
>
> Summarized results:
>
> |Method | top-1 acc. (%)  | GFlops |
> | ---------- | -------------------  | ---------- |
> | LDAM+DRW                                                     | 42.0              | 1$\times$ |
> | Ensemble of two LDAM+DRW models [1]  | 45.0              | 2$\times$ |
> | RIDE w/  $L_{\text{Diversity}}$                    | 45.0               | 1$\times$ |
> | RIDE w/  $L_{\text{D-Diversity}}$ [2]           | 46.6              | 1.1$\times$ |
> | RIDE w/  $L_{\text{D-Diversity}}$ w/ self-distill w/EA [3]  | 47.0    | 0.9$\times$ |
> | Difference between [2] and [1]                     |  1.6           | -0.9$\times$ |
> | Difference between [3] and [1]                     |  2.0           | -1.1$\times$ |
>
> While our distribution-aware diversity loss $L_{\text{D-Diversity}}$ gets higher performance gains with 3 or more experts especially for few-shot classes, RIDE (the [3] in the table above) still gains by 2% against the ensemble method and by more than 5% against the baseline method with only 2 experts and at a much smaller compute.

---

> > ### Author Response · Authors · 2020-11-23
> > **Thanks so much for your constructive and encouraging feedback! (Part 2/3)**
> >
> > **[Q2]** The expert routing/assignment module dynamically routes cascaded experts, and all the passed experts before the stop will jointly make a decision. How to make sure the early experts are the more robust ones? Since experts have already been frozen when the routing module is trained, and the diversity loss won't force the early experts to be more robust. There is no guarantee that the involved early experts are more useful than the skipped ones.
> >
> > **[A2]** That's a great question!  Our model deploys experts in a sequence at the test time, although these experts are not specially selected during training.  We test 3 methods for expert assignment (EA) on a 4-expert model on ImageNet-LT, comparing 1) no selection where all 4 experts are used in parallel, 2) random selection where all 4 experts are chosen simply in their nominal order, 3) optimal selection where the 4 experts are chosen in the order of their overall accuracies on the validation set.
> >
> > |Setting       | Overall acc. (%) | Many, Medium, Few acc. (%) |
> > | --- | --- | --- |
> > |4 Experts (No EA, All Used)   | 46.702            | 58.56, 42.60, 27.95|
> > |4 Experts (Random EA)          | 46.562            | 58.47, 42.44, 27.74|
> > |4 Experts (Optimal EA)           | 46.584            | 58.46, 42.49, 27.78 |
> >
> > The performance achieved by the first setting where all 4 experts are used on all the instances could be regarded as an upper bound for the expert assignment module.  These results show that: 1) The performance gap caused by various expert re-orderings is very small, all close to the upper bound; 2) Ordering the experts based on the overall validation accuracy results in a statistically insignificant 0.02% increase.  In addition, we look into the pass rate of each expert assignment module on the test set: It differs only by less than 5% between random and optimal ordering of experts, which means that there is practically no GFlops difference between the two settings.
> >
> > These results in addition to Figures 4 and 5 in the paper help clarify the key aspect of our approach in terms of reducing the variance towards correct predictions.  1) All our experts are trained simultaneously to optimize the classification accuracy while producing more diversity for classes with fewer instances.  Consequently, they are comparable overall, with likewise higher/lower performance for head/tail instances.   2) Our router is trained not to directly increase the accuracy of a collective decision, but to reduce the uncertainty towards a correct prediction.  For more difficult tail instances, more and diverse experts are used, effectively reducing the variance of their collective decision.  For easier head instances, fewer experts are used; with each at a higher accuracy, the higher variance associated with fewer experts does not impact the correctness of their collective decision.  3) While it is possible to further optimize the performance by ordering and tuning experts, since the individual strengths of the experts is not what our method relies on, the room for improvement is likely to be small.

---

> > > ### Author Response · Authors · 2020-11-23
> > > **Thanks so much for your constructive and encouraging feedback! (Part 3/3)**
> > >
> > > **[Q3]** The experts in RIDE may learn to be biased toward different categories due to the diversity loss, and the routing module works as an outlier detector which only stops when it finds a proper expert biased toward the underline ground-truth labels. Is it another way to understand RIDE? or do I misunderstand some parts of our paper?
> > >
> > > **[A3]** Another great question!  To continue our discussion on the similarities and differences between individual experts, we examine more difficult tail-class instances.  We find out that there are cases where each expert may classify them wrong, but their collective decision turns out right.  The table below lists the softmax probabilities for two such examples:  The correct class is never the winning prediction by each expert, but it has a consistent medium-sized response; when aggregated together, it becomes the eventual winner in the collective decision.
> > >
> > > |Choices |     Expert 1                      |      Expert 2           |      Expert 3          | Joint prediction |
> > > |---|---|---|---|---|
> > > |First   | steel arch bridge (50%)    | steel arch bridge (57%) | pier (84%)             | crane (45%, correct)|
> > > |Second  | crane (42%) (correct)   | crane (21%) (correct)   | crane (11%, correct)   | steel arch bridge (31%)|
> > > |Third   | paddlewheel (3%)            | pier (20%)              | drilling platform (2%) | pier (20%)|
> > >
> > > |Choices |     Expert 1                       |      Expert 2           |      Expert 3          | Joint prediction |
> > > |---|---|---|---|---|
> > > |First   | burrito (33%)                      | dough (37%)             | French loaf (49%)      | potpie (36%, correct)|
> > > |Second  | French loaf (25%)          | potpie (29%, correct)   | potpie (19%, correct)  | dough (20%)|
> > > |Third   | potpie (20%, correct)       | bagel (16%)             | burrito (18%)          | French loaf (14%)|
> > >
> > > These results are consistent with the key of our algorithm on reducing the variance towards correct predictions.  As our router is trained to optimize the collective decision, it is not so much about a subsequent expert getting the prediction right, but about the reduction of prediction variance in the correct class with the accumulation of diverse expert opinions. The head-class instances can be reliably predicted by individual (and thus fewer) experts, whereas tail-class instances can only be predicted by individual experts with so much uncertainty that the consistency among diverse experts becomes a crucial performance booster.
> > >
> > > We appreciate these questions that deepen our understanding of RIDE.  We will include such analysis and discussions in the revision.

---

### Official Review · AnonReviewer4 · 2020-10-26
**In this paper, authors propose a stage-wise multi-expert system for long-tailed recognition problem.**

**Rating:** 7
**Confidence:** 4

**Review:**

The majority of feature extraction backbone is shared among different agents and the classifiers of experts are trained with both classification loss and proposed distribution-aware diversity loss. For the second stage, an expert assignment module is trained to re-weight the expert decisions. The whole paper is generally well-organized. However, there are some technical issues authors should further address:
1) In the introduction, the authors measure the mean accuracy, model bias and model variance by randomly sampling the Cifar100 a few times, and reported different methods in Fig1. The details of how such bias and variance is computed are not given. For example, it is stated the bias also measures the accuracy.
2) For the motivation of distribution-aware diversity loss, if the same input batch is observed by all the experts, why is it 'distribution aware'? Also, why requires the diversity among experts if setting the lower temperature to tail classes is to encourage the divergent prediction, how both diversity and divergent exists simultaneously?
3) Is the input batch randomly sampled from the entire training set?
4) For experiments on ImageNet-LT and iNaturalist, is the RIDE combined with the regular CE loss or other methods?
5) Can authors demonstrate some details regarding the performance of expert assignment module during the test, as this is the core module in the proposed framework? For example, for each split (many, low) how the different expert behaves?

[Post Rebuttal Comments] Authors have done a good job for addressing my concerns, especially the additional ablation studies regarding the performance of the expert assignment module. I'm updating my score accordingly for recommending acceptance.

---

> ### Author Response · Authors · 2020-11-23
> **Thanks so much for your detailed questions and constructive comments. (Part 1/2)**
>
> Thanks so much for your detailed questions and constructive comments. We address each of your comments below.
>
> **[Q1]** In the introduction, the authors measure the mean accuracy, model bias and model variance by randomly sampling the Cifar100 a few times, and reported different methods in Fig1. The details of how such bias and variance is computed are not given. For example, it is stated the bias also measures the accuracy.
>
> **[A1]** The error of a regression model can be decomposed as $Err(x) = \text{Bias}^2 + \text{Variance} + \sigma^2_e$, where $\sigma^2_e$ is the irreducible error, which represents an absolute lower bound.  We follow the procedure in *A unified bias-variance decomposition* by Pedro Domingos to empirically compute such a decomposition for a classification model. Below is a quick summary.
>
> Let $D$ be a set of training sets, $x$ be a sample in the dataset.  A loss function $\mathcal{L}(t; y)$ measures the cost of predicting $y \in Y$ when the true value is $t$. Assume the optimal prediction $y*$ for $x$ is the prediction that minimizes $E_t[\mathcal{L}(t; y*)]$, where the subscript $t$ denotes that the expectation is taken with respect to all possible values of $t$, weighted by their probabilities given $x$.  The main prediction $y_m$ for a loss function $\mathcal{L}$ over the  training set $D$ (which is treated as a random variable) is defined as $y_m=\text{argmin}_{y'}E_D[L(y,y')]$.  That is, the main prediction is the one that "differs least" from all the predictions in $Y$ according to $\mathcal{L}$.
>
> The bias is the loss incurred by the main prediction $y_m$ relative to the optimal prediction $y*$, and the variance is the average loss incurred by predictions $y$ relative to the main prediction $y_m$. We then evaluate bias and variance with zero one loss according to the evaluation metric in "A unified bias-variance decomposition" [1] for classification, i.e. $\text{bias} = \mathcal{L}_\text{zero-one}(y*, y_m), \text{variance}=E_D[\mathcal{L}_\text{zero-one}(y_m, y)]$.
>
> **[Q2]** For the motivation of distribution-aware diversity loss, if the same input batch is observed by all the experts, why is it 'distribution aware'?
>
> **[A2]** How we use the data distribution is different from previous works, which use the distribution for  re-balancing data in the training procedure or re-weighting training instances in the loss function.  In our work, we use the distribution to regulate the degree of inter-expert correlation per class in the loss function; our loss desires more diverse experts for classes with fewer training instances.
>
> With re-weighting which gives more weight to tail classes, each expert is trained to improve the performance more on tail-classes, consequently their predictions have higher variances due to small numbers of training instances.  With our distribution aware diversity loss, we are not asking each expert to focus more on some specific instances; instead, we are asking a multi-expert system to deliver more diversified decisions on tail-class instances.
>
> **[Q3]** Also, why requires the diversity among experts if setting the lower temperature to tail classes is to encourage the divergent prediction, how both diversity and divergent exists simultaneously?
>
> **[A3]** Sorry about the confusion! The lower temperature T, the larger the softmax of $\Delta x$ would make in the prediction.  The temperature controls the sensitivity to the difference in the feature space.  The lower the temperature, the larger the sensitivity.  The head-class instances tend to have a larger neighbourhood in the feature space, whereas the tail-class instances have a small neighbourhood in the feature space.  Having the temperature reflect such a natural divergent data distribution in the same feature space would facilitate learning of the classifier weight parameters, whereas the diversity among experts is captured by the KL divergence in the loss function.
>
> **[Q4]** Is the input batch randomly sampled from the entire training set?
>
> **[A4]** Yes, for each iteration, we randomly sample a mini-batch of instances from the entire training set.
>
> **[Q5]** For experiments on ImageNet-LT and iNaturalist, is the RIDE combined with the regular CE loss or other methods?
>
> **[A5]** Yes, it is combined with LDAM, which is proposed specifically for long-tailed data classification. Figure 3 in the paper shows that RIDE combined with CE already delivers good performance on CIFAR100-LT. We select LDAM by default for its simplicity and strong performance. Two-stage methods such as cRT and $\tau-$norm require an additional training stage. RIDE also applies to these methods and can also deliver a performance boost to these approaches.
>
> [1] Domingos, Pedro. "A unified bias-variance decomposition." In Proceedings of 17th International Conference on Machine Learning, pp. 231-238. 2000.

---

> > ### Author Response · Authors · 2020-11-23
> > **Thanks so much for your detailed questions and constructive comments. (Part 2/2)**
> >
> > **[Q6]** Can authors demonstrate some details regarding the performance of expert assignment module during the test, as this is the core module in the proposed framework? For example, for each split (many, low) how the different expert behaves?
> >
> > **[A6]** Yes!  Figure 4 in the paper shows the number of experts versus top-1 accuracy for each split (All, Many/Medium/Few) of CIFAR100-LT. Figure 5 in the paper shows the proportion of the number of experts allocated to each split of CIFAR100-LT.
> >
> > In addition, we conduct some ablation studies on CIFAR100-LT to further explore how the different experts behave on each split.  The results are listed in the three tables below.  The first table shows that performance difference between different experts is greatest for the few-shot classes.  The second table shows that the experts are widely different overall.  The third table shows that for all the experts perform similarly at a high level of accuracies on the first-round instances (easy cases), and more differently at a low level of accuracies on the last-round instances (difficult cases).  More details are provided under each table.
> >
> > |Setting      | Overall acc. (%)  | Many, Medium, Few acc. (%)|
> > |---|---|---|
> > |4 Experts (without EA**)   | 49.3              | 69.20, 49.65, 26.32|
> > |4 Experts (with EA)   | 49.1              | 69.34, 49.26, 26.00|
> > |Expert 1     | 44.0^             | 63.89, 44.5^, 21.10|
> > |Expert 2     | 44.1              | 63.06^,46.68*,19.58^|
> > |Expert 3     | 44.6*             | 63.74, 45.15, 22.39*|
> > |Expert 4     | 44.2              | 64.69*,44.74, 20.58|
> > |max - min     |  0.6              |  1.63,  2.18,  2.81|
> >
> > \^: worst-performing expert; *: best-performing expert; **: Without EA means making use of all experts on all instances without expert assignment module (EA), which is an approximate upper bound for expert assignment module.
> >
> > According to the table, the difference between best performing and worst performing expert on few-shot classes can reach about 2.8\% and the overall accuracy can differ in about 0.6\%. Our expert assignment model recognizes the individual characteristics of each expert and works nearly optimally in choosing experts, as demonstrated in the negligible difference (0.2\%, which is similar to the level of overall training randomness) in accuracy with and without Expert Assignment module (first two experiments in the table).
> >
> > We also conducted experiments with RIDE (4 experts) on CIFAR100-LT to statistically quantify the diversity between each pair of the 4 experts, as listed below. Each item (row $i$, column $j$) is the degree of agreement between expert $i$ and $j$ (i.e. the percentage of test set instances that two experts obtain the same prediction).
> >
> > |       |Expert 1| Expert 2| Expert 3 |Expert 4|
> > |--|--|--|--|--|
> > |**Expert 1** |100.  |  53.17 |  53.34| 53.18|
> > |**Expert 2** | 53.17|  100.   | 52.77|  53.38|
> > |**Expert 3** | 53.34|   52.77| 100.  |  52.87|
> > |**Expert 4** | 53.18|   53.38|  52.87| 100.  |
> >
> > This table shows that our experts only have about half predictions in common, which indicates the diversity among predictions from various experts.
> >
> > The dynamic expert assignment (EA) module and the framework with diverse experts are a symbiotic system, the collaboration between the two components delivers the significant performance improvements to state-of-the-arts on long-tailed distribution datasets. To further understand the effectiveness of EA module, we then conduct a controlled experiment: In all the samples allocated to only one expert, we intentionally add more experts cumulatively to investigate the changes in performance when more experts participate; similarly, we calculate the accuracy over samples allocated to all 4 experts.
> >
> > |  Experts used | Accuracy over samples assigned to only Expert 1 (%)  | Accuracy over samples assigned to all 4 experts (%) |
> > | --- | --- | --- |
> > |Expert 1       | 72.39                          | 21.39|
> > |Experts 1-2    | 72.60                          | 25.03|
> > |Experts 1-3    | 72.70                          | 28.46|
> > |Experts 1-4    | 72.81                          | 29.01|
> > |max - min      |  0.42                          |  7.62|
> >
> > We found that the samples assigned to only one expert by EA module are mostly easy samples (accuracies are around 72\%), recruiting more experts can only obtain statistical negligible performance improvements. The ones that are allocated to all 4 experts are mostly hard samples (accuracies are less than 30\%), assigning more experts leads to about 7.6\% accuracy improvements. Therefore, this table proves that EA module can effectively allocate an appropriate number of experts to each instance.
> >
> > Hope our explanation and experiments are able to address your inquiries. Please don't hesitate to reply if you have any further concerns.

---

### Official Review · AnonReviewer3 · 2020-10-29
**This paper proposed a simple but effective method to solve long-tailed recognition. It finds out that current methods suffer high bias and variance. To relieve this problem, it propose to ensemble several experts to make predictions. It proposes a diversiy loss to guarantee the diversity among experts and learn an expert assignment module to turn on/off an expert when predicting. The proposed method of this paper is general and significantly outperforms the state-of-the-art method by 5% to 7%.**

**Rating:** 7
**Confidence:** 4

**Review:**

This paper proposed a simple but effective method which significantly outperforms the state-of-the-art method by 5% to 7%. The experiments are adequate and rigorous. Besides these , the writing is clear and easy to understand. Totally, this paper is a good work
Pros:
1. The writing is clear.
2. The view of bias and variance is novel and interesting.
3. The experiments are adequate and rigorous.
4. The performance of proposed method is very strong.
5. The proposed method is simple, effective and general.
Cons:
1. The procedure of test is not clear enough. For example, how to turn on/off an expert? By a threshold?
2. The description of the part about shared, indepent, and recuded extractor is not clear enough.

---

> ### Author Response · Authors · 2020-11-23
> **Thank you for your constructive and encouraging feedback!**
>
> We would like to thank you for your constructive and encouraging feedback. Here is our response to your comments:
>
> **[Q1]** The procedure of test is not clear enough. For example, how to turn on/off an expert? By a threshold?
>
> **[A1]** Sorry about the brevity of our test procedure! We have added more descriptions. Basically, with a dynamic expert routing module, RIDE assigns another trained and distinctive expert for a second (or third, ...)  opinion when it is called for. Since we use a binary cross entropy loss with logits input during training, the output of this module is logits that decide whether we need extra activations from more experts. The predicted probability can be obtained by appending a sigmoid function at the test time. In our implementation, if the predicted probability is greater than or equal to threshold 0.5, we decide to add another expert. Another EA module will be used to decide whether yet another expert is needed until we use up all the experts we have.
>
> This threshold is a hyper-parameter, which could be adjusted to "to turn on/off an expert", or more precisely adjust the proportion of samples passing through each expert, or control the "easiness for each expert assignment module to be satisfied". In our current implementation though, we fix this threshold and adjust the positive weight of the binary cross entropy loss (i.e., $\omega_p$ in Eqn 5) instead. The latter hyper-parameter only exists during training; by increasing the weight $\omega_p$, training would focus more on samples that require more experts and thus boost their performance. With our EA module dynamically routing experts (saving unnecessary expert computation), the computational complexity is greatly reduced, at only negligible 0.3\% reduction in the overall accuracy.
>
> **[Q2]** The description of the part about shared, independent, and recuded extractor is not clear enough.
>
> **[A2]** Thanks for giving feedback about the extractor description! Here we clarify the description of our shared feature extractor and independent parts of individual experts.  Let's use ResNet as an example backbone to illustrate our concept. We divide ResNet into two parts: one part with the first two stages and one part with the last two stages along with the fully connected layer. Then we duplicate the second part N times, where N is the number of experts.  The first part is the shared extractor while the second part is independent among individual experts. During both training and evaluation times, as illustrated in Fig. 2a, we pass an image to the first part only once, which produces an intermediate representation; we then pass the intermediate representation to each expert. In the end, after getting the logits, we take the arithmetic mean of logits of all the experts and pass the logits to softmax to get the probability. A similar procedure could be applied to other models such as ResNeXt. By the way, we assume that "recuded" refers to "reduced". If this is not correct, feel free to follow up!
>
> In order to reduce the computational complexity, instead of using the full module, we reduce the number of channels in the convolutional layers of each expert by 1/4 and adjust other layers accordingly. We do not have a specific reduced extractor.

---

### Official Review · AnonReviewer1 · 2020-10-29
**Official Blind Review #1**

**Rating:** 8
**Confidence:** 5

**Review:**

This paper proposes a method termed RoutIng Diverse Experts (RIDE) for reducing both the bias and the variance of a long-tailed classifier. Specifically, RIDE consists of three crucial components: 1) a shared architecture for multiple experts; 2) a distribution-aware diversity loss that encourages more diverse decisions for classes with fewer training instances; 3) an expert routing module that dynamically assigns more ambiguous instances to additional experts. Experiments are conducted on three long-tailed benchmark datasets, i.e., CIFAR100-LT, ImageNet-LT, and iNaturalist. Satisfactory classification results of long-tailed visual recognition are observed.

Paper strengths:
- The problem, i.e., long-tailed visual recognition, is practical, important and challenging in computer vision and deserves further studies.
- The proposed method has good motivations and sounds reasonable.
- The experimental results of the proposed RIDE are significantly better than the results of previous work, which shows the effectiveness of RIDE.
- The paper is well written and easy to follow.
- Analyses and ablation studies are sufficient and could bring new insights of long-tailed recognition.

Paper weaknesses:
- There are typos and grammar mistakes in this paper. For example, in the caption of Fig. 2, "ImaeNet-LT and BBN" should be "ImageNet-LT and BBN". The authors should carefully proofread the final version.
- Some references have inconsistent reference formats, e.g, "In Proceedings of the IEEE conference on computer vision
and pattern recognition" of [Xie et al., 2017] vs. "In Proceedings of the IEEE/CVF Conference on Computer Vision and Pattern Recognition" of [Zhou et al., 2020] vs. "CVPR09" of [Deng et al., 2009]. In addition, some other references lack detailed information, e.g., [He and Garcia, 2009] lacks page, volume information. Some references also have capital issues, e.g., "The inaturalist species classification" -> "The iNaturalist species classification" of [Van Horn et al., 2018], and "Bbn" -> "BBN" of [Zhou et al., 2020].

---

> ### Author Response · Authors · 2020-11-23
> **Thanks so much for the positive confirmation of our work!**
>
> Thanks so much for the positive confirmation of our work!
>
> **[Q1]** There are typos and grammar mistakes in this paper. The authors should carefully proofread the final version. Some references have inconsistent reference formats.
>
> **[A1]** Thank you for your thorough review. We have updated the paper to address your concerns on the typos and reference format mismatches.

---

### Author Response · Authors · 2020-11-23
**We thank reviewers for their insightful and overall positive comments.**

We thank reviewers for their insightful and overall positive comments. R1 finds our experimental results "significantly better than the results of previous work, which shows the effectiveness". R2 considers our work "a new direction for long-tailed classification", and that it "firstly increases the performances on all three splits (many-/med-/few-shot), while most of the existing methods have to sacrifice the head for tail improvements". R3 believes that "the view of bias and variance is novel and interesting" and "the proposed method is simple, effective and general." R4 considers our paper "generally well-organized." We are committed to public code release and result reproducibility.

For the rebuttal, we make several changes: 1. A detailed explanation of bias and variance, including equations and calculation methods, is added in the introduction. 2. Typos and reference format mismatches are corrected. 3. We add more clarifications on the expert assignment module. 4. We add more clarifications on the overall multi-expert framework.

---

### Decision · Program_Chairs · 2021-01-07
**Final Decision**

**Decision:**

Accept (Spotlight)

**Comment:**

The paper addresses the important problem of classification with unbalanced semantic classes. The key idea is a two-stage process that first learns a representation under various distributions (experts), then assign a cascade of experts to hard samples, whose predictions are combined. The approach can be added on top of various backbone networks. Experiments are systematic and extensive, showing improvement on three standard benchmarks for this task.

All four reviewers recommended accept.  The paper extends a recent research direction of learning "balanced" representations, followed by distribution-aware experts. This general approach could have wider impact on architectures designed for out-of-distribution and low-shot learning.  The authors should update the final paper based on their answers and on reviewer feedback. We also encourage authors to make their code available.